# SAAL: SHARPNESS-AWARE ACTIVE LEARNING

## ABSTRACT

While modern deep neural networks play significant roles in many research areas, they are also prone to overfitting problems under limited data instances. Particularly, this overfitting, or generalization issue, could be a problem in the framework of *active learning* because it selects a few data instances for learning over time. To consider the generalization, this paper introduces the first active learning method to incorporate the sharpness of loss space in the design of the acquisition function, inspired by sharpness-aware minimization (SAM). SAM intends to maximally perturb the training dataset, so the optimization can be led to a flat minima, which is known to have better generalization ability. Specifically, our active learning, Sharpness-Aware Active Learning (SAAL), constructs its acquisition function by selecting unlabeled instances whose perturbed loss becomes maximum. Over the adaptation of SAM into SAAL, we design a pseudo labeling mechanism to look forward to the perturbed loss w.r.t. the ground-truth label. Furthermore, we present a theoretic analysis between SAAL and recent active learning methods, so the recent works could be reduced to SAAL under a specific condition. We conduct experiments on various benchmark datasets for vision-based tasks in image classification and object detection. The experimental results confirm that SAAL outperforms the baselines by selecting instances that have the potentially maximal perturbation on the loss.

## 1 INTRODUCTION

Recently, deep learning is widely utilized in many research areas, such as computer vision, natural language processing, recommender systems, etc., but its success deeply depends on the large-scale labeled dataset for training the deep neural networks. The importance of the dataset is related to the generalization issue in deep learning, which refers that the model learned with the training dataset suffers from the degradation of performance when the unseen test dataset is encountered for deployment. This degradation results from the neural networks that are prone to overfitting under the lack of the training dataset (Keskar et al., 2016; Neyshabur et al., 2017; Kawaguchi et al., 2017).

The dependency on the dataset also invokes an adaptive data selection by acquisition functions, or active learning, which aims at the efficient use of the limited budget for annotations from oracle (Cohn et al., 1996; Tong, 2001; Settles, 2009). Recently, various methods for active learning have been proposed; but the model trained with a small number of data from the adaptive selection is often difficult to be generalized (Dasgupta & Hsu, 2008). Although there exist some prior works that deal with the generalization issue in active learning; those methods solve the problem by either proposing a new risk function (Farquhar et al., 2020) or adopting a new classifier network (Wan et al., 2021), rather than by inventing a new acquisition function that considers the generalization.

In this paper, we propose a new active learning algorithm, named Sharpness-Aware Active Learning (SAAL), that connects active learning and generalization ability to construct the acquisition function. Specifically, we are inspired by Sharpness-Aware Minimization, or SAM (Foret et al., 2020), which minimizes the maximally perturbed loss of training dataset, leading to minimizing the loss sharpness as well as the task loss, itself. Such optimization leads to a flat minima of the loss landscape, which is shown to have a strong correlation with the generalization performance (Jiang et al., 2019). Hence, SAAL adopts the maximally perturbed loss as the acquisition score.

When calculating the acquisition score for SAAL, we cannot observe the labels for the unlabeled instances, so it is infeasible to compute the perturbed loss. To overcome this challenge, we utilize

pseudo labels predicted by the current model, and we theoretically show that our proposed pseudo labeling conservatively estimates the maximally perturbed loss w.r.t. ground-truth label. Also, we theoretically derive the upper bound of the acquisition score of SAAL, which includes the loss, the norm of gradients, and the first eigenvalue of loss Hessian. Among the three terms of the upper bound, the loss and gradient terms are widely used metrics for active learning, which captures the model change by acquiring the instance (Yoo & Kweon, 2019; Ash et al., 2020; Settles et al., 2007). Meanwhile, the first eigenvalue, which is newly considered by SAAL, is connected to the loss sharpness (Keskar et al., 2017). Therefore, the selected instances by SAAL might contribute to the generalization of the model.

We summarize our contributions in three points. First, we propose Sharpness-Aware Active Learning (SAAL), which considers the loss sharpness for constructing the acquisition function. The loss sharpness is related to the generalization of model, so selecting instances with a high value of loss sharpness might lead to a model with a better generalization performance. Second, we theoretically derive the upper bound of the acquisition score of SAAL and show the connection with the recent active learning methods. Specifically, we find that the upper bound also contains the first eigenvalue of loss Hessian, which is related to the generalization ability. Third, we empirically show that SAAL outperforms the baselines in various vision-based tasks on the benchmark dataset.

## 2 PRELIMINARIES

### 2.1 NOTATIONS

Throughout this paper, we assume a classification problem and we represent our current deep learning model parameterized by $\theta$ as $f_\theta \colon \mathbb{R}^d \to \mathbb{R}^{|\mathscr{Y}|}$; where $d$ is the dimension of data instance, $x$, and $\mathscr{Y}$ is the set of candidate classes that $x$ can have. There are two datasets: a dataset with labels, $\mathscr{X}_L$, and the other unlabeled dataset, $\mathscr{X}_U$. We denote the acquisition function of active learning as $f_{acq} \colon \mathbb{R}^d \to \mathbb{R}$, where $f_{acq}$ receives a data instance as input, and calculates the informativeness, or the acquisition score, of the instance as output. The loss of a data instance, $x$, w.r.t. the given label $y$ is represented as $l(x, y; \theta) \coloneqq l_{\text{CE}}(\sigma(f_\theta(x)), y)$, where $\sigma(\cdot)$ is a softmax function. The total loss of a dataset, $S$, is represented as $L_S(\theta) = \frac{1}{N} \sum_{i=1}^{N} l(x_i, y_i; \theta)$, where $S = \{(x_i, y_i) | i = 1, ..., N\}$. Lastly, we define the pseudo label, $\hat{y} = \text{argmax}_{j \in \mathscr{Y}} \sigma(f_\theta(x))_j$; and we denote the ground-truth label as $\bar{y}$.

### 2.2 ACTIVE LEARNING

There are several active learning scenarios that differ by the setting of data accessibility; which include membership-query synthesis (Angluin, 1988; 2004), stream-based active learning (Atlas et al., 1989; Cohn et al., 1994), and pool-based active learning (Lewis & Gale, 1994). In this paper, we focus on pool-based active learning, where the unlabeled data instances are provided as a large set of data pool, and the active learner sequentially selects the informative instances by a certain criterion.

Pool-based active learning is categorized by the definition of informativeness, which includes the uncertainty, diversity, and hybrid-based methods. **Uncertainty-based active learning** adopts the acquisition function, $f_{acq}$, to calculate the uncertainty of each unlabeled instance with regard to the current deep learning model, and an oracle provides the ground-truth label of the selected unlabeled instances with the highest uncertainty. Since the acquisition score is usually calculated for an unlabeled instance, $x_u \in \mathscr{X}_U$, w.r.t. the current model, $f_\theta$, it is expanded as $f_{acq}(x_u; f_\theta)$, resulting in the selection rule as the below.

$$\mathscr{X}_S = \underset{\mathscr{X}_S' \subset \mathscr{X}_U}{\text{argmax}} \sum_{x_u \in \mathscr{X}_S'} f_{acq}(x_u; f_\theta) \tag{1}$$

Entropy, which is denoted as $f_{acq}^{Ent}(x_u; f_\theta) = \mathbb{H}[f_\theta(x_u)] = -\sum_j \sigma(f_\theta(x_u))_j \log_2 \sigma(f_\theta(x_u))_j$, or variation ratio, which is denoted as $f_{acq}^{Var} = 1 - \max_j \sigma(f_\theta(x_u))_j$, are the most widely used methods for calculating uncertainty (Shannon, 1948; Freeman, 1965). These days, additional networks are used to approximate the uncertainty of each instance. Learning Loss for Active Learning (LL4AL) (Yoo & Kweon, 2019) trains the loss prediction module, $f_{LPM}$, which takes the hidden feature maps

as input and predicts the expected loss as output. Then, LL4AL constructs the acquisition functions $f_{acq}^{LL4AL}(x_u) = f_{LPM}(f_\theta^k(x_u)|_{k=1,...,K})$, where $f_\theta^k$ is the $k$-th hidden feature map. Variational Adversarial Active Learning (VAAL) (Sinha et al., 2019) trains a discriminator, $f_{dis}$, which takes a data instance as input and discriminates whether the instance belongs to the labeled dataset or the unlabeled dataset. Then, VAAL calculates the probability of $x_u$ belonging to the unlabeled dataset, $\mathcal{X}_U$, as the acquisition score, i.e., $f_{acq}^{VAAL}(\mathcal{X}_U) = f_{dis}(x_u)$. **Diversity-based active learning**, such as Coreset approach (Sener & Savarese, 2018), selects instances that represent the whole distribution of unlabeled instances, by solving a mixed integer programming. Recently, to make use of both uncertainty and diversity methods, **Hybrid-based active learning** is proposed to select the uncertain instances in a diverse way. In BADGE (Ash et al., 2020), the acquisition function is calculated as the gradient embedding of $x_u$ w.r.t. the parameter of the last fully connected layer, $\theta_{out}$, that is $f_{acq}^{BADGE}(x_u) = \frac{\partial}{\partial \theta_{out}} l(x_u, \hat{y}_u; \theta)$, where $\hat{y}_u$ is the pseudo label of $x_u$. Then, this embedding becomes an input to the k-means++ seeding algorithm (Arthur & Vassilvitskii, 2006).

Recently, pool-based active learning has been developed to deal with certain problematic scenarios. Such development includes using random round-robin sampling to efficiently apply active learning in large-batch setting (Citovsky et al., 2021); actively selecting test samples to query (Kossen et al., 2021); improving conventional active learning algorithm for high dimensional observational data (Jesson et al., 2021); etc. Compared to those researches, our proposed SAAL deals with a conventional scenario and focuses on how the proposed acquisition function selects the informative instances.

## 2.3 SHARPNESS-AWARE MINIMIZATION (SAM)

As an independent research direction from active learning, there is an increasing investigation on the flatness or sharpness of loss response surfaces, and their corresponding optimization. The flat minima of loss landscape is analyzed in various research areas and is confirmed to have deep connection to generalization of neural networks. A recent broad study on the various measures of generalization has confirmed that sharpness-based measure has the strongest correlation to the generalization (Jiang et al., 2019). Hence, the flat minima is utilized in various research areas where generalization is important, such as domain generalization (Cha et al., 2021), adversarial robustness (Stutz et al., 2021), or domain adversarial training (Rangwani et al., 2022).

Having said that, Sharpness-Aware Minimization (SAM) is an optimizer for training the deep neural network (Foret et al., 2020) to weigh the importance of flat minima. Denoting the loss on the dataset $S$ w.r.t. the current parameter $\theta$ as $L_S(\theta)$, the optimization objective of SAM is to minimize the maximally perturbed loss with the regularization on the parameter, as below.

$$\min_\theta \max_{\|\epsilon\| \leq \rho} L_S(\theta + \epsilon) + \gamma \|\theta\|_2^2 \tag{2}$$

Here, $\gamma$ is a hyperparameter that controls the magnitude of the effect of regularization, $\epsilon$ is the perturbation to the parameter, and $\rho$ defines the size of the perturbation.

The maximally perturbed loss can be decomposed as $\max_{\|\epsilon\| \leq \rho} L_S(\theta + \epsilon) = (\max_{\|\epsilon\| \leq \rho} L_S(\theta + \epsilon) - L_S(\theta)) + L_S(\theta)$, which is interpreted as the sharpness term (first term of the RHS) and the classification loss term (second term of the RHS). Hence, SAM minimizes the sharpness of the loss as well as the classification loss value, itself; that is SAM aims at seeking the flat minima among the local minimas. This optimization is a max-min problem, which solves $\max_{\|\epsilon\| \leq \rho} L_S(\theta + \epsilon)$ first than solves $\min_\theta \max_{\|\epsilon\| \leq \rho} L_S(\theta + \epsilon)$. The inner maximization problem is solved by finding $\epsilon^* = \text{argmax}_{\|\epsilon\| \leq \rho} L_S(\theta + \epsilon)$. By deriving Taylor expansion of $L_S(\theta + \epsilon)$ w.r.t. $\theta$ around 0, and by introducing a dual norm problem, the $\epsilon^*$ is approximated as the follow, with $\frac{1}{p} + \frac{1}{q} = 1$.

$$\epsilon^* \approx \rho \cdot \text{sign}(\nabla_\theta L_S(\theta)) \frac{|\nabla_\theta L_S(\theta)|^{q-1}}{(\|\nabla_\theta L_S(\theta)\|_q^q)^{1/p}} \tag{3}$$

After solving the inner maximization using $\epsilon^*$, the minimization problem is solved by obtaining the gradient, while excluding the Hessian term, as the below.

$$\nabla_\theta \max_{\|\epsilon\| \leq \rho} L_S(\theta + \epsilon) \approx \nabla_\theta L_S(\theta)|_{\theta + \epsilon^*} \tag{4}$$

## 3 METHOD

### 3.1 MOTIVATION

According to SAM (Foret et al., 2020), the loss of the population dataset, $\mathscr{D}$, is upper bounded by the maximally perturbed loss of the training dataset, $\mathscr{X}$. From the perspective of active learning, the training dataset is decomposed into the labeled dataset, $\mathscr{X}_L$, and the unlabeled dataset, $\mathscr{X}_U$, i.e., $\mathscr{X} = \mathscr{X}_L \cup \mathscr{X}_U$. Hence, the upper bound can be decomposed as the below, with $\pi_L = \frac{|\mathscr{X}_L|}{|\mathscr{X}|}$ and $\pi_U = \frac{|\mathscr{X}_U|}{|\mathscr{X}|}$.

$$L_{\mathscr{D}}(\theta) \leq \max_{\|\epsilon\| \leq \rho} L_{\mathscr{X}}(\theta + \epsilon) \leq \pi_L \cdot \max_{\|\epsilon\| \leq \rho} L_{\mathscr{X}_L}(\theta + \epsilon) + \pi_U \cdot \max_{\|\epsilon\| \leq \rho} L_{\mathscr{X}_U}(\theta + \epsilon) =: L_{\mathscr{X}}^{SAAL} \quad (5)$$

Since the population loss, $L_{\mathscr{D}}(\theta)$, is never accessible; we instead access the rightmost upper bound denoted in Eq. 5, which is represented as $L_{\mathscr{X}}^{SAAL}$, and train our network to minimize the upper bound. Among the two terms of $L_{\mathscr{X}}^{SAAL}$, the first term, $\pi_L \cdot \max_{\|\epsilon\| \leq \rho} L_{\mathscr{X}_L}(\theta + \epsilon)$, will be minimized if we use SAM optimizer. Then, the remaining second term, $\pi_U \cdot \max_{\|\epsilon\| \leq \rho} L_{\mathscr{X}_U}(\theta + \epsilon)$, becomes the key component for our optimization in the sharpness-aware active learning scenario. During the active learning iterations, we suppose that we select unlabeled instances, $x_u \in \mathscr{X}_U$, with high values of maximally perturbed loss. In other words, we query the label of such unlabeled instances, so that the remaining $\mathscr{X}_U$ consists of unlabeled instances whose maximally perturbed loss value is small. Then, it leads to minimizing two terms of $L_{\mathscr{X}}^{SAAL}$ simultaneously; which contributes to minimizing the generalization error, $L_{\mathscr{D}}(\theta)$.

**Comparison to Semi-Supervised Learning**    Our proposed active learning algorithm is not the only way for decreasing the loss of unlabeled dataset, $\mathscr{X}_U$; because traditional semi-supervised learning (SSL) is another way that utilizes $L_{\mathscr{X}_U}(\theta)$ while training the model. However, it should be noted that SSL does not guarantee to minimize the rightmost upper bound, $L_{\mathscr{X}}^{SAAL}$. SSL minimizes the average of unlabeled dataset loss instead of the maximum perturbed loss. Hence, it is hard to guarantee that SSL will contribute to minimizing the generalization error without prior knowledge on label distribution (Ben-David et al., 2008). We can categorize the SSL approach as three ways (Berthelot et al., 2019; Zhu, 2005), which are consistency regularization (Laine & Aila, 2016; Sajjadi et al., 2016), entropy minimization (Cireşan et al., 2010; Lee et al., 2013), and traditional regularization, such as weight decay (Zhang et al., 2018a;b). First, consistency regularization and entropy minimization completely depend on the pseudo-label, and an incorrect pseudo-label might increase the generalization error. Second, the worst-case or hardest instances might have incorrect pseudo-label. In other words, SSL, training the model with an incorrect pseudo-label, might fail to model the maximum perturbed loss. Third, the minimization of maximum perturbed loss is an independent approach to the previous semi-supervised learning methods, such as traditional regularization as well as consistency and entropy minimization. They are potentially compatible with our active learning algorithm.

### 3.2 SHARPNESS-AWARE ACTIVE LEARNING

Motivated by the Sharpness-Aware Minimization (SAM), our active learning algorithm selects instances with a high perturbed loss under some perturbation on the model parameters, $\theta$. Hence, our acquisition function is as follows:

$$f_{acq}^{SAAL}(x_u; f_\theta) = \max_{\|\epsilon\| \leq \rho} l(x_u, \hat{y}_u; \theta + \epsilon), \quad (6)$$

where $l$ is the cross-entropy loss function for the model, and $\theta$ is the current parameter of the model. Algorithm 1 describes the overall process of our Sharpness-Aware Active Learning. Since our acquisition function is calculated for the unlabeled instances, there comes a problem when calculating the maximally perturbed loss function, which requires label. Hence, we use a pseudo label, $\hat{y}_u$, for the loss calculation. To provide the validity of utilizing pseudo label, we first provide Theorem 3.1, which explains the relation between the maximally perturbed losses which are calculated with pseudo label and with ground-truth label, respectively.

**Theorem 3.1.** *(Proof in Appendix A.2.1) For a data instance $x$, let $\hat{y}$ be the pseudo label predicted by the network $f_\theta$ and $\bar{y}$ be the ground-truth label. Then, the maximally perturbed loss calculated with*

---

**Algorithm 1** Sharpness-Aware Active Learning

---

**Input:** Labeled dataset $\mathscr{X}_L^0$, Unlabeled dataset $\mathscr{X}_U^0$, Classifier $f_\theta$
1: Initially train $f_\theta$ by the cross-entropy loss of $\mathscr{X}_L^0$
2: **for** $j = 0, 1, 2, \ldots$ **do** $\qquad\qquad\qquad\qquad\qquad\qquad\qquad\qquad$ ▷ active learning
3: $\qquad$ Randomly sample $\mathscr{X}_U^{pool} \subset \mathscr{X}_U^j$
4: $\qquad$ **for** $x_u \in \mathscr{X}_U^{pool}$ **do**
5: $\qquad\qquad$ Calculate $f_{acq}^{SAAL}(x_u; f_\theta)$ as Eq. 6
6: $\qquad$ **end for**
7: $\qquad$ Select $\mathscr{X}_S = \text{argmax}_{\mathscr{X}_S' \subset \mathscr{X}_U^{pool}} \sum_{x_u \in \mathscr{X}_S'} f_{acq}^{SAAL}(x_u; f_\theta)$
8: $\qquad$ Query the label of $\mathscr{X}_S$ to oracle
9: $\qquad$ Update the labeled dataset, $\mathscr{X}_L^{j+1} = \mathscr{X}_L^j \cup \mathscr{X}_S$
10: $\qquad$ Update the unlabeled dataset, $\mathscr{X}_U^{j+1} = \mathscr{X}_U^j \setminus \mathscr{X}_S$
11: $\qquad$ Train $f_\theta$ by the cross-entropy loss of $\mathscr{X}_L^{j+1}$
12: **end for**

---

$(x, \hat{y})$ *is a lower bound of the maximally perturbed loss calculated with* $(x, \bar{y})$; *with a non-negative margin,* $\delta_x$, *as the below:*

$$\max_{\|\epsilon\| \leq \rho} l(x, \hat{y}; \theta + \epsilon) \leq \max_{\|\epsilon\| \leq \rho} l(x, \bar{y}; \theta + \epsilon) + \delta_x. \tag{7}$$

Next, Proposition 3.2 shows that the inequality 7 has zero margin under a mild condition.

**Proposition 3.2.** *(Proof in Appendix A.2.2) For a data instance $x$ and the corresponding pseudo label $\hat{y}$, let $\hat{\epsilon}$ be the maximal perturbation over the parameters w.r.t. the loss $l(x, \hat{y}; \theta + \epsilon)$. If the perturbed network, $f_{\theta+\hat{\epsilon}}$, keeps the predicted label as the same as the label predicted from the original network, $f_\theta$; then the maximally perturbed loss calculated with $(x, \hat{y})$ is a lower bound of the maximally perturbed loss calculated with $(x, \bar{y})$, as the below:*

$$\max_{\|\epsilon\| \leq \rho} l(x, \hat{y}; \theta + \epsilon) \leq \max_{\|\epsilon\| \leq \rho} l(x, \bar{y}; \theta + \epsilon). \tag{8}$$

Theorem 3.1 and Proposition 3.2 provide that the selected instances by acquisition score of SAAL with pseudo label would also have high scores w.r.t. the ground-truth label. It indicates that we conservatively estimate the maximally perturbed loss for the acquisition score.

### 3.3 CONNECTION TO RECENT ACTIVE LEARNING ALGORITHMS

Here, we theoretically derive the upper bound of the acquisition score of SAAL, and show the connection to the recent active learning algorithms as well as the generalization ability. To begin with, we provide Theorem 3.3 as below.

**Theorem 3.3.** *(Proof in Appendix A.2.3) The acquisition function, $f_{acq}^{SAAL}$, of Eq. 6 is upper bounded by $l(\theta) + \rho\|\nabla_\theta l(\theta)\|_2 + \frac{1}{2}\rho^2\lambda_1 + \max_{\|v\| \leq 1} O(\rho^2 v^3)$; where $l(\theta)$ abbreviates the loss of a data pair, $(x, y)$, and $\lambda_1$ is the first eigenvalue of the loss Hessian.*

Theorem 3.3 derives the upper bound of the acquisition score of SAAL, which consists of the task loss, the gradient norm, and the first eigenvalue of loss Hessian. Since we are selecting instances which have high value of $f_{acq}^{SAAL}$, the selection refers that we are also selecting instances which have high values of the loss, $l(\theta)$, and the magnitude of the gradient embedding, $\|\nabla_\theta l(\theta)\|_2$; which are connected to LL4AL (Yoo & Kweon, 2019) and BADGE (Ash et al., 2020), respectively. Furthermore, SAAL considers the first eigenvalue of the loss Hessian w.r.t. the current model, denoted as $\lambda_1$. The importance of the first eigenvalue for generalization is widely studied, that is the first eigenvalue is used as the indicator of the sharpness of the loss surface (Keskar et al., 2017; Zhuang et al., 2022; Kaur et al., 2022). Hence, the selected instances by SAAL might contribute to the generalization of the model.

Figure 1a shows that there exists a positive correlation between our acquisition score, $f_{acq}^{SAAL}$, and the three terms of upper bound. It should be noted that the upper bound is not our optimization

objective, because we use the acquisition score for ranking the instances with top values of the score. Hence, the correlation between the acquisition score and the upper bound is of our interest. Having said that, by selecting the instances with the high acquisition score of SAAL, $f_{acq}^{SAAL}$, we are selecting instances that have high values of the loss, gradient norm, and the first eigenvalue. Also, Figure 1b shows the value of the three terms of upper bound. Interestingly, as the acquisition iterations proceed, not only the loss and the gradient value, but the first eigenvalue gets smaller. The change of the value of the first eigenvalue is more noticeable in Figure 1c, which plots the value of $\lambda_1$ without the scaling term of $\frac{1}{2}\rho^2$. This indicates that SAAL leads the model to a flat minima, which results in better generalization performance.

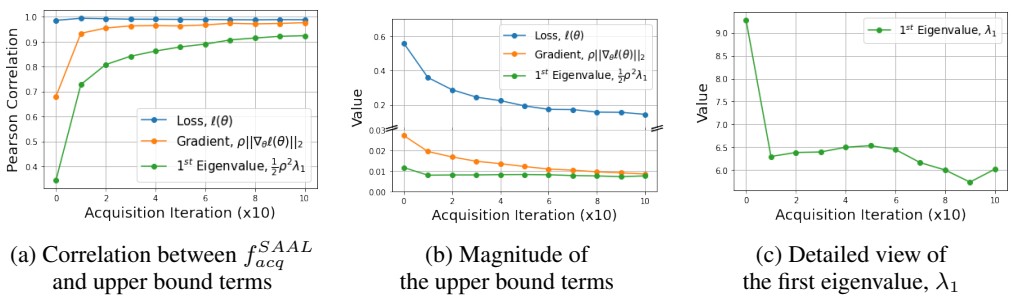

(a) Correlation between $f_{acq}^{SAAL}$ and upper bound terms

(b) Magnitude of the upper bound terms

(c) Detailed view of the first eigenvalue, $\lambda_1$

Figure 1: Correlation and magnitude of upper bound terms

## 4 EXPERIMENTS AND RESULTS

We examined the performance of SAAL in two vision-based tasks; which are image classification and object detection. Specifically, for image classification, we quantitatively compared the test accuracies among various active learning algorithms; and qualitatively analyzed the performance of SAAL by examining the upper bound of population loss. Also, we analyze the effect of $\rho$, which defines the size of the perturbation, $\epsilon$.

### 4.1 IMAGE CLASSIFICATION

**Experiment Setting**   We conduct our experiment on Fashion-MNIST (Fashion) (Xiao et al., 2017), SVHN (Netzer et al., 2011), CIFAR-10, and CIFAR-100 (Krizhevsky et al., 2009) dataset. We adopt the ResNet-18 (He et al., 2016) network for our classifier, and we train the network for 50 epochs after each acquisition step, using Adam optimizer (Kingma & Ba, 2015) with a learning rate of 1e-3; and SAM optimizer (Foret et al., 2020) with a learning rate of 1e-3 for Fashion, SVHN, CIFAR-10 and 1e-1 for CIFAR-100. To simulate an experiment scenario with bad generalization cases, we followed the settings of the prior works (Kim et al., 2021), which assumes a very low amount of allowed budget. For Fashion, SVHN, and CIFAR-10, we construct the initial labeled dataset with 20 instances, which are random but balanced; and we select 10 additional instances with the highest acquisition score among the randomly selected 2,000 unlabeled instances. For CIFAR-100, the initial labeled dataset consists of 1,000 instances, and we select 100 additional instances for 100 repeated iterations. We repeat the acquisition for 100 iterations, and we report the results with three repeated trials. Here, SAAL introduces the perturbation size, $\rho$, of the perturbation, $\epsilon$, in Eq. 6, and we set the value of $\rho$ as 0.05 for all the datasets.

**Baselines**   We compared the performance of SAAL with Random, Entropy (Shannon, 1948), Coreset (Sener & Savarese, 2018), Learning Loss for Active Learning (LL4AL) (Yoo & Kweon, 2019), Variational Adversarial Active Learning (VAAL) (Sinha et al., 2019), and BADGE (Ash et al., 2020). Since our most relevant algorithm, BADGE, adopts k-means++ seeding algorithm to introduce diversity on the acquisition; we also provide an experimental result with diversity following the same practice from BADGE. Specifically, after calculating our acquisition function using Eq. 6, we implement k-means++ seeding algorithm with the acquisition score as input.

**Quantitative Analysis**   Table 1 indicates that SAAL outperforms the baselines in seven out of eight combinations of experiments. The advantage of SAAL becomes obvious when we use the Adam optimizer, rather than the SAM optimizer. We conjecture that this gain for Adam optimizer

originates from Eq. 5, which motivates SAAL in modeling the expected flat local minima after acquisitions. Recall that our inaccessible goal, $L_{\mathscr{D}}(\theta)$, is upper bounded by $\pi_L \cdot \max_{\|\epsilon\| \leq \rho} L_{\mathscr{X}_L}(\theta + \epsilon) + \pi_U \cdot \max_{\|\epsilon\| \leq \rho} L_{\mathscr{X}_U}(\theta + \epsilon)$, as we discussed in Section 3.1. When using Adam optimizer, the first term, $\max_{\|\epsilon\| \leq \rho} L_{\mathscr{X}_L}(\theta + \epsilon)$, in the upper bound is weakly optimized compared to using SAM optimizer, which we will present qualitative analyses in the next section; because SAM optimizer directly minimizes $\max_{\|\epsilon\| \leq \rho} L_{\mathscr{X}_L}(\theta + \epsilon)$. Hence, the importance of the second term in the upper bound, $\max_{\|\epsilon\| \leq \rho} L_{\mathscr{X}_U}(\theta + \epsilon)$, becomes more significant for Adam optimizer.

We also provide the test accuracy along the acquisition iterations in Figure 6 of Appendix A.1. The figures show that SAAL achieves higher accuracy more quickly than baselines in most cases, see Figure 6a, 6d, or 6g.

Table 1: Comparison of test accuracy (%) using Adam optimizer and SAM optimizer. The best performance is indicated as boldface. The results are replicated by three times.

| Method | Adam optimizer | | | | SAM optimizer | | | |
|---|---|---|---|---|---|---|---|---|
| | Fashion | SVHN | CIFAR-10 | CIFAR-100 | Fashion | SVHN | CIFAR-10 | CIFAR-100 |
| Random | $81.2 \pm 0.5$ | $72.4 \pm 0.9$ | $50.7 \pm 1.5$ | $43.3 \pm 0.3$ | $83.7 \pm 0.3$ | $78.1 \pm 1.1$ | $52.6 \pm 2.8$ | $44.0 \pm 0.7$ |
| Entropy | $81.5 \pm 1.4$ | $73.1 \pm 1.0$ | $51.9 \pm 1.8$ | $44.4 \pm 0.7$ | $84.1 \pm 0.2$ | $77.5 \pm 3.2$ | $54.6 \pm 0.4$ | $44.1 \pm 1.0$ |
| Coreset | $83.8 \pm 0.7$ | $75.3 \pm 5.8$ | $51.7 \pm 1.0$ | $44.4 \pm 0.5$ | $84.4 \pm 0.6$ | $\mathbf{78.9 \pm 1.3}$ | $53.9 \pm 1.3$ | $47.6 \pm 1.4$ |
| LL4AL [1] | $83.5 \pm 1.8$ | $75.1 \pm 1.7$ | $51.7 \pm 0.4$ | $43.9 \pm 0.3$ | $83.2 \pm 1.4$ | $72.2 \pm 0.2$ | $50.2 \pm 1.1$ | $35.7 \pm .01$ |
| VAAL | $83.4 \pm 0.1$ | $73.4 \pm 1.3$ | $52.0 \pm 0.9$ | $44.8 \pm 0.3$ | $84.1 \pm 0.6$ | $77.1 \pm 0.8$ | $53.1 \pm 0.9$ | $45.5 \pm 0.4$ |
| BADGE | $85.4 \pm 0.6$ | $74.9 \pm 1.1$ | $52.3 \pm 2.2$ | $45.7 \pm 0.6$ | $86.2 \pm 0.2$ | $78.8 \pm 0.9$ | $56.8 \pm 1.9$ | $47.4 \pm 0.7$ |
| SAAL | $\mathbf{85.8 \pm 0.8}$ | $\mathbf{76.8 \pm 0.7}$ | $\mathbf{54.4 \pm 0.9}$ | $\mathbf{47.6 \pm 0.9}$ | $\mathbf{86.3 \pm 0.5}$ | $78.8 \pm 1.0$ | $\mathbf{57.0 \pm 1.1}$ | $\mathbf{48.4 \pm 0.9}$ |

Next, we compare the time complexity of SAAL and baselines. We used the CIFAR-10 dataset and measured the time for a single iteration of acquisition and training. Figure 2 shows the time in seconds with a log scale. The results of Random acquisition show that the SAM optimizer takes twice longer time than the Adam optimizer, because it takes two steps of gradient calculation. However, the gap between Adam and SAM becomes mere when using other active learning algorithms, indicating that the time for calculating acquisition score is the largest bottleneck. SAAL calculates the perturbation, $\epsilon$, for every single unlabeled instance, instead of batch-wise calculation; so it takes longer than most of the other baselines. The time complexity of SAAL can be reduced if we adopt the improved SAM models (Du et al., 2021; 2022) that have been proposed for an efficient calculation.

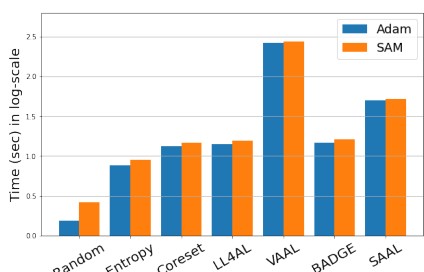

Figure 2: Comparison of time complexity, in log-scale

**Qualitative Analysis** Figure 3 supports the conjecture for the advantage of SAAL by anticipating the flat local minima in the acquisition process. Figure 3 measures the maximally perturbed loss for the labeled dataset, $\mathscr{X}_L$; the unlabeled dataset, $\mathscr{X}_U$; and the total dataset, $\mathscr{X}_L \cup \mathscr{X}_U$. We compare the results between the models trained with either SAM optimizer or Adam optimizer. Since it is computationally hard to calculate the corresponding perturbation for every single unlabeled instance, $x_u \in \mathscr{X}_U$, we uniformly sample 2,000 unlabeled instances from $\mathscr{X}_U$ at each iteration; and we report the averaged results for three independently repeated trials.

Figure 3a shows the maximally perturbed loss of $\mathscr{X}_U$ when using SAM optimizer. If we compare the result of SAAL with the results of baselines, SAAL shows the lowest value of the maximally perturbed loss, because SAAL selected the instances with high values of perturbed loss; and because SAAL removes such instances by passing those instances to the labeled dataset. Figure 3b shows the maximally perturbed loss of $\mathscr{X}_L$ when using SAM optimizer. This loss also indicates the flatness of the model; the lower value of the maximally perturbed loss of $\mathscr{X}_L$ indicates that the model does not change the result even if the parameter is changed in a small range, which refers to the flat model (Keskar et al., 2016; Neyshabur et al., 2017; Kawaguchi et al., 2017). Hence, SAAL results in a flat network compared to the baselines. We conjecture that the flat model attained by SAAL is explained

---

[1]For LL4AL, we failed to reproduce the performance when using SAM optimizer.

by the look-ahead concept (Roy & McCallum, 2001; Konyushkova et al., 2017; Kim et al., 2021). If we are planning to minimize $\max_{\|\epsilon\| \leq \rho} L_{\mathscr{X}}(\theta + \epsilon)$ by SAM optimizer, SAAL looks ahead the high values of the $\max_{\|\epsilon\| \leq \rho} L_{\mathscr{X}}(\theta + \epsilon)$ from unlabeled instances, and SAAL actively selects such unlabeled instances to flatten the future response surface.

Finally, Figure 3c shows the maximally perturbed loss of the total dataset, which is equivalent to the upper bound in Eq. 5. As confirmed in the figure, SAAL achieves the lowest upper bound, which indicates that the model trained with SAAL is more likely to achieve a lower population loss, which is our ultimate goal of minimization objective. When comparing the results of SAM (Figure 3a - Figure 3c) and the results of Adam (Figure 3d - Figure 3f), the gap between SAAL and other baselines becomes clearer in Adam optimizer.

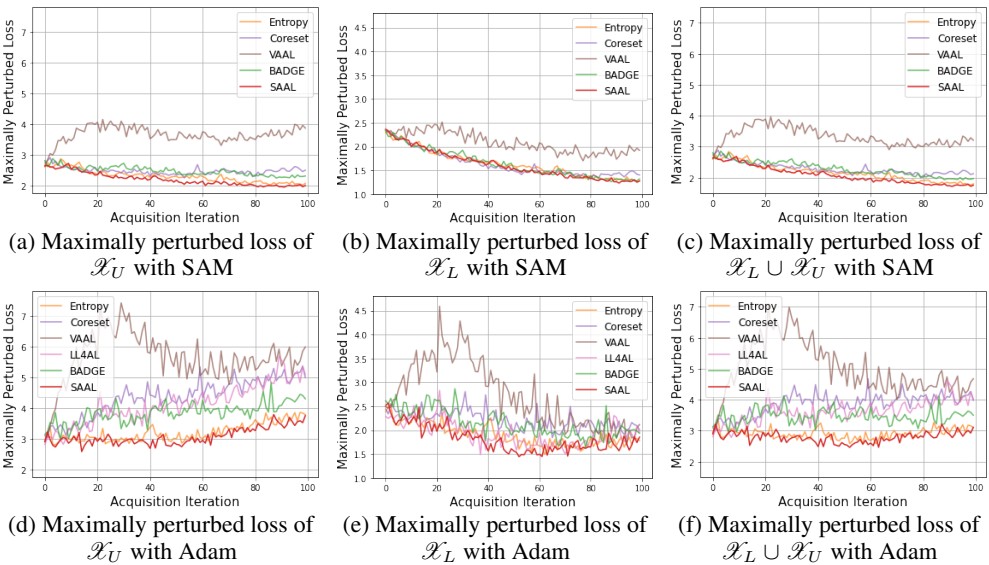

(a) Maximally perturbed loss of $\mathscr{X}_U$ with SAM

(b) Maximally perturbed loss of $\mathscr{X}_L$ with SAM

(c) Maximally perturbed loss of $\mathscr{X}_L \cup \mathscr{X}_U$ with SAM

(d) Maximally perturbed loss of $\mathscr{X}_U$ with Adam

(e) Maximally perturbed loss of $\mathscr{X}_L$ with Adam

(f) Maximally perturbed loss of $\mathscr{X}_L \cup \mathscr{X}_U$ with Adam

Figure 3: Maximally perturbed loss of the labeled dataset, unlabeled dataset, and total dataset during the active learning iterations. (a) - (c) are the results of the model trained by SAM optimizer. (d) - (f) are the results of the model trained by Adam optimizer.

**Sensitivity Analysis on $\rho$** SAAL introduces a hyperparameter, $\rho$, which represents the size of the perturbation, $\epsilon$. Hence, we conduct the sensitivity analysis on $\rho$ with the CIFAR-10 dataset, and we set the candidate values for $\rho$ as 0.01, 0.05, and 0.10.

First, we examined the validity of Theorem 3.1 by investigating if the network with the maximally perturbed parameters keeps the predicted label as same as the original network. Figure 4a shows the proportion of unlabeled data instances whose predicted labels are remained the same by the perturbed network during the active learning iterations; that is $\psi(\rho) :=$ $\frac{1}{|\mathscr{X}_U|} \sum_{x \in \mathscr{X}_U} \mathbf{1}_{\operatorname{argmax}_j f_{\theta+\hat{\epsilon}}(x)_j = \hat{y}}$, where $\mathbf{1}_A$ is the indicator function that returns 1 if the condition $A$ is satisfied. When the size, $\rho$, of the perturbation, $\epsilon$, is zero, that is if we do not perturb the network; then the inequality of Eq. 8 is satisfied for all instances, by the definition of the pseudo label, $\hat{y}$. As we increase the value of $\rho$, some instances fail to keep the predicted label as the same as $\hat{y}$, because the parameter of the model changes drastically so that the model loses the prediction ability that it has learned so far. Also, we examined the validity of Proposition 3.2 by investigating the value of the margin, $\delta_x$, for the unlabeled data instances in Figure 4b. It should be noted that $\delta_x$ is not our hyperparameter, but a dependent variable subject to change by $\rho$. We only investigate $\delta_x$ to reveal the characteristics of $\rho$, not for the hyperparameter optimizations. To show how the value of the margin, $\delta_x$, affects the inequality, we measure the relative value of the margin, $\delta_x$, compared to the maximally perturbed loss, $\max_{\|\epsilon\| \leq \rho} l(x, \bar{y}; \theta + \epsilon)$; that is $r(\rho, \delta_x) := \frac{1}{|\mathscr{X}_U|} \sum_{x \in \mathscr{X}_U} \frac{\delta_x}{\max_{\|\epsilon\| \leq \rho} l(x, \bar{y}; \theta + \epsilon)}$. From the analyses of Figure 4a and 4b, we adopted $\rho = 0.05$, because this value 1) keeps the predicted label of data instance from the original network with high probability and 2) keeps the value of the margin relatively small compared the max perturbed loss w.r.t. the ground-truth label; while $\rho = 0.05$ is confirmed to perturb the parameters of the network effectively (Foret et al., 2020).

The proper selection of $\rho$ also affects the test accuracy, as shown in Figure 4c. If we select $\rho$ with a too small value, that is $\rho = 0.01$, the parameter of the model is not perturbed enough to measure the sharpness, so SAAL cannot catch the informative instances. If we select $\rho$ with a too large value, that is $\rho = 0.10$, the maximally perturbed loss 1) does not satisfy Proposition 3.2, as confirmed in Figure 4a, and 2) have too large value of margin, as confirmed in Figure 4b. Meanwhile, a proper value of $\rho = 0.05$ for the perturbation, $\epsilon$, shows the best performance.

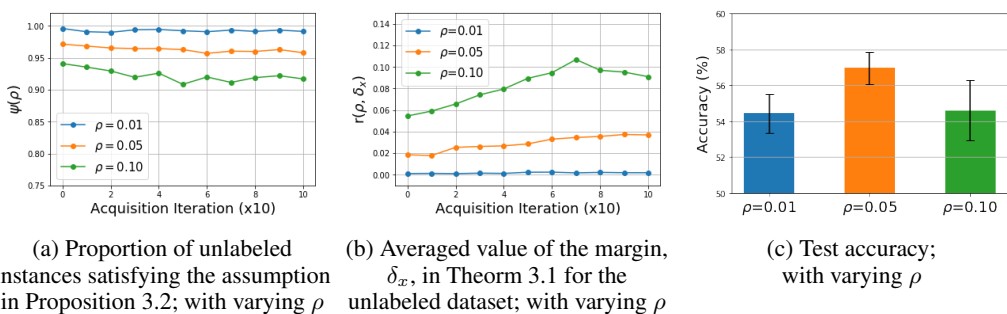

(a) Proportion of unlabeled instances satisfying the assumption in Proposition 3.2; with varying $\rho$

(b) Averaged value of the margin, $\delta_x$, in Theorm 3.1 for the unlabeled dataset; with varying $\rho$

(c) Test accuracy; with varying $\rho$

Figure 4: Sensitivity analysis on $\rho$

## 4.2 OBJECT DETECTION

To show the effectiveness of SAAL in a complex task, we conduct an object detection task. Object detection returns the locations of semantic objects and the corresponding labels for a given input image, $x$. Hence, the loss for training detection model consists of the bounding box regression loss and the classification loss. We experiment with PASCAL VOC 2007 and 2012 dataset (Everingham et al., 2010), which contains 5,011 images and 4,952 images with 20 object classes, respectively. We adopt Single Shot Multibox Detector (SSD) (Liu et al., 2016) as the detection model. To apply SAAL for object detection, we perturb the parameters to maximize the classification loss; and use the summation of the perturbed loss from every corresponding detection box in the image, $x$, as the acquisition score for $x$. Afterward, we select the images with the highest scores. We construct the initial labeled dataset with 1,000 randomly selected images, and we select additional 1,000 instances at every acquisition iterations, so that we attain 10,000 final instances with nine repeated acquisitions. We train the model for 300 epochs with a batch size of 32. Figure 5 reports the mean average precision (mAP) for three repeated trials of SAAL and baselines. As shown in the figure, SAAL achieves high performance at the earlier iterations and shows the highest mAP of 0.7541 at the last iteration; while BADGE, Entropy, and Random show 0.7493, 0.7518, and 0.7403, respectively.

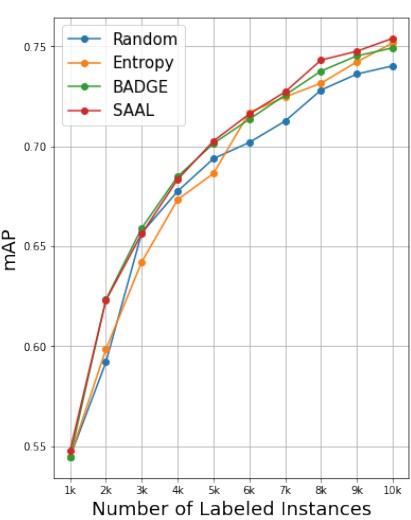

Figure 5: mAP of object detection task with PASCAL VOC 2007+2012

## 5 CONCLUSION AND FUTURE WORKS

We propose a new active learning method named Sharpness-Aware Active Learning, or SAAL. SAAL considers the loss sharpness of data instances, which is strongly related to the generalization performance of deep learning. Furthermore, we derive the upper bound of SAAL acquisition score and find the connection to the recent active learning methods; as well as the connection to the first eigenvalue of loss Hessian which is widely used as the indicator of loss sharpness. By various experiments with benchmark datasets, SAAL shows better performance than baselines. As a future work, we will improve the time complexity of SAAL by adopting recent SAM models.

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

# A  APPENDIX

## A.1  TEST ACCURACY

We provide the learning curve of SAAL and baselines along the acquisition iterations.

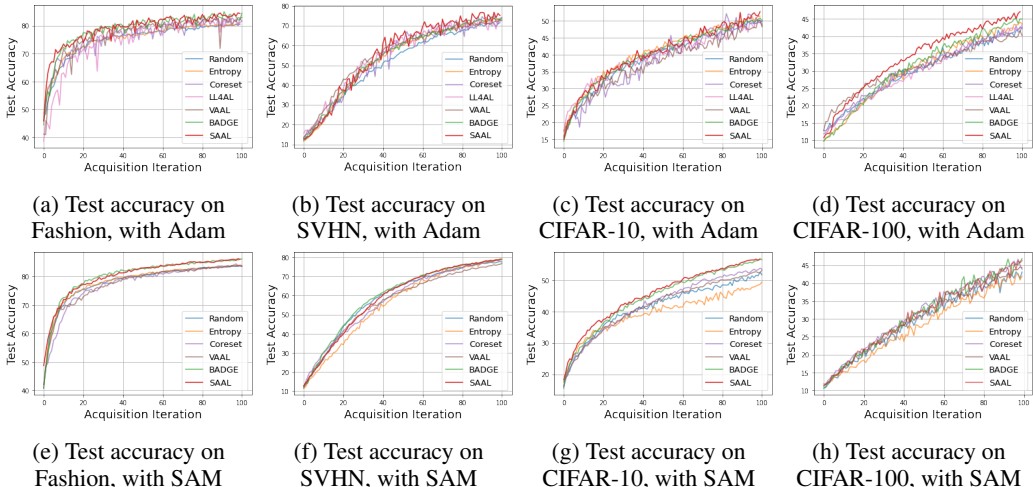

(a) Test accuracy on Fashion, with Adam

(b) Test accuracy on SVHN, with Adam

(c) Test accuracy on CIFAR-10, with Adam

(d) Test accuracy on CIFAR-100, with Adam

(e) Test accuracy on Fashion, with SAM

(f) Test accuracy on SVHN, with SAM

(g) Test accuracy on CIFAR-10, with SAM

(h) Test accuracy on CIFAR-100, with SAM

Figure 6: Test accuracy along the acquisition iteration; with Adam and SAM optimizers

## A.2  PROOF DETAILS

### A.2.1  PROOF OF THEOREM 3.1

**Theorem A.1.** *For a data instance $x$, let $\hat{y}$ be the pseudo label predicted by the network $f_\theta$ and $\bar{y}$ be the ground-truth label. Then, the maximally perturbed loss calculated with $(x, \hat{y})$ is a lower bound of the maximally perturbed loss calculated with $(x, \bar{y})$; with a non-negative margin, $\delta_x$, as the below:*

$$\max_{\|\epsilon\| \leq \rho} l(x, \hat{y}; \theta + \epsilon) \leq \max_{\|\epsilon\| \leq \rho} l(x, \bar{y}; \theta + \epsilon) + \delta_x.$$

*Proof.* The cross-entropy loss, $l(x, y; \theta)$, is represented with the logit vector $f_\theta(x) \in \mathbb{R}^{|\mathcal{Y}|}$ as the below:

$$l(x, y; \theta) = -\ln \frac{\exp(f_\theta(x)_y)}{\sum_j \exp(f_\theta(x)_j)}$$
$$= -\ln \left(\exp(f_\theta(x)_y)\right) + \ln \sum_j \exp(f_\theta(x)_j)$$
$$= \ln \sum_j \exp(f_\theta(x)_j) - f_\theta(x)_y.$$

Then, the maximally perturbed loss of a data pair $(x, y)$ is represented as the below:

$$\max_{\|\epsilon\| \leq \rho} l(x, y; \theta + \epsilon) = \max_{\|\epsilon\| \leq \rho} (\ln \sum_j \exp(f_{\theta+\epsilon}(x)_j) - f_{\theta+\epsilon}(x)_y).$$

Since the pseudo label, $\hat{y}$, satisfies $\hat{y} = \text{argmax}_{j \in \mathcal{Y}} f_\theta(x)_j$ by the definition, it holds that $f_\theta(x)_{\hat{y}} \geq f_\theta(x)_j$ for all $j \in \mathcal{Y}$. Let $\hat{\epsilon} = \text{argmax}_{\|\epsilon\| \leq \rho} l(x, \hat{y}; \theta + \epsilon)$. Define the margin, $\delta_x$, as $\delta_x :=$

$[\max_j\{f_{\theta+\hat{\epsilon}}(x)_j - f_{\theta+\hat{\epsilon}}(x)_{\hat{y}}\}]_+$ where $[\cdot]_+ = \max\{\cdot, 0\}$. Then, the following holds.

$$
\begin{aligned}
\max_{\|\epsilon\|\le\rho} l(x, \hat{y}; \theta + \epsilon) &= \ln \sum_j \exp(f_{\theta+\hat{\epsilon}}(x)_j) - f_{\theta+\hat{\epsilon}}(x)_{\hat{y}} \\
&\le \ln \sum_j \exp(f_{\theta+\hat{\epsilon}}(x)_j) - f_{\theta+\hat{\epsilon}}(x)_{\bar{y}} + \delta_x \\
&\le \max_{\|\epsilon\|\le\rho} \left( \ln \sum_j \exp(f_{\theta+\epsilon}(x)_j) - f_{\theta+\epsilon}(x)_{\bar{y}} \right) + \delta_x \\
&= \max_{\|\epsilon\|\le\rho} l(x, \bar{y}; \theta + \epsilon) + \delta_x
\end{aligned}
$$

$\square$

### A.2.2 PROOF OF PROPOSITION 3.2

**Proposition A.2.** *For a data instance $x$ and the corresponding pseudo label $\hat{y}$, let $\hat{\epsilon}$ be the maximal perturbation over the parameters w.r.t. the loss $l(x, \hat{y}; \theta + \epsilon)$. If the perturbed network, $f_{\theta+\hat{\epsilon}}$, keeps the predicted label as the same as the label predicted from the original network, $f_\theta$; then the maximally perturbed loss calculated with $(x, \hat{y})$ is a lower bound of the maximally perturbed loss calculated with $(x, \bar{y})$, as the below:*

$$
\max_{\|\epsilon\|\le\rho} l(x, \hat{y}; \theta + \epsilon) \le \max_{\|\epsilon\|\le\rho} l(x, \bar{y}; \theta + \epsilon).
$$

*Proof.* Since the perturbed network, $f_{\theta+\hat{\epsilon}}$, keeps the predicted label as the same as the label predicted from the original network, $f_\theta$; it holds that $\operatorname{argmax} f_{\theta+\hat{\epsilon}}(x) = \operatorname{argmax} f_\theta(x) = \hat{y}$ and accordingly $f_{\theta+\hat{\epsilon}}(x)_j \le f_{\theta+\hat{\epsilon}}(x)_{\hat{y}}$ for all $j$. Hence, $\max_j\{f_{\theta+\hat{\epsilon}}(x)_j - f_{\theta+\hat{\epsilon}}(x)_{\hat{y}}\} \le 0$. Thus, by the definition of the margin in Theorem 3.1, $\delta_x$ becomes zero.

$\square$

### A.2.3 PROOF OF THEOREM 3.3

**Theorem A.3.** *The acquisition function, $f_{acq}^{SAAL}$, of Eq. 6 is upper bounded by $l(\theta) + \rho\|\nabla_\theta l(\theta)\|_2 + \frac{1}{2}\rho^2\lambda_1 + \max_{\|v\|\le1} O(\rho^2 v^3)$; where $l(\theta)$ abbreviates the loss of a data pair, $(x, y)$, and $\lambda_1$ is the first eigenvalue of the loss Hessian.*

*Proof.* Recall that our acquisition function is $f_{acq}^{SAAL} = \max_{\|\epsilon\|\le\rho} l(x_u, \hat{y}_u; \theta + \epsilon)$. Since we limit the size of the perturbation as $\|\epsilon\| \le \rho$, we can write $\epsilon = \rho v$ with $\|v\| \le 1$, and $\max_{\|\epsilon\|\le\rho} l(x_u, \hat{y}_u; \theta + \epsilon) = \max_{\|\rho v\|\le\rho} l(x_u, \hat{y}_u; \theta + \rho v) = \max_{\|v\|\le1} l(x_u, \hat{y}_u; \theta + \rho v)$. Then, by Taylor expansion of $l(x_u, \hat{y}_u; \theta + \rho v)$ w.r.t. $\theta$, the below holds, where we abbreviate $l(x_u, \hat{y}_u; \theta)$ as $l(\theta)$.

$$
\begin{aligned}
f_{acq}^{SAAL}(x_u; f_\theta) &= \max_{\|\epsilon\|\le\rho} l(\theta + \epsilon) = \max_{\|v\|\le1} l(\theta + \rho v) \\
&= \max_{\|v\|\le1} \{ l(\theta) + (\rho v)^T \nabla_\theta l(\theta) + \frac{1}{2}(\rho v)^T \nabla_\theta^2 l(\theta)(\rho v) + O((\rho v)^3) \} \\
&= l(\theta) + \max_{\|v\|\le1} \{ (\rho v)^T \nabla_\theta l(\theta) + \frac{1}{2}(\rho v)^T \nabla_\theta^2 l(\theta)(\rho v) + O((\rho v)^3) \} \\
&\le l(\theta) + \max_{\|v\|\le1} (\rho v)^T \nabla_\theta l(\theta) + \max_{\|v\|\le1} \frac{1}{2}(\rho v)^T \nabla_\theta^2 l(\theta)(\rho v) + \max_{\|v\|\le1} O((\rho v)^3) \\
&= \underbrace{l(\theta)}_{\text{Loss}} + \underbrace{\rho\|\nabla_\theta l(\theta)\|_2}_{\text{Gradient Norm}} + \underbrace{\frac{1}{2}\rho^2\lambda_1}_{1^{st}\text{ Eigenvalue}} + \max_{\|v\|\le1} O((\rho v)^3)
\end{aligned}
$$

$\square$

