# OpenReview forum: "SAAL: Sharpness-Aware Active Learning"
_ICLR.cc/2023/Conference — Submitted to ICLR 2023_

### Official Review · Reviewer_Xi2g · 2022-10-23

**Confidence:** 4
**Correctness:** 3
**Technical Novelty And Significance:** 2
**Empirical Novelty And Significance:** 2
**Recommendation:** 5

**Clarity, Quality, Novelty And Reproducibility:**

Quality: The proposed method is simple and makes sense. The performance is promising. The proposed method is inspired based on SAM by taking the generalization ability into account for active learning. But it may need more innovations to improve the performance.
Clarity: The paper is well written. Easy to read and follow.
Originality: The work introduces the loss sharpness from SAM into active learning to improve the generalization ability. However, it doesn’t provide more universal generalization improvement method that matches better with adaptive learning setting.


**Strength And Weaknesses:**


Strength:
a. The approach is simple. It is interesting to see that selecting instances with a high value of loss sharpness improves the performance.
b. The results look promising.
c. The manuscript is easy to read

Weaknesses
a.	The motivation may overlook the relationship of the selected instances and unselected ones. The work simply reduces the second term in Eq.5 by removing the unlabeled instances with highest loss sharpness. The challenges to compute sharpness should be further explained.

b.	The method to estimate the loss sharpness of the unlabeled instances may be easy. And a wide lower bound in Theorem 3.1 may be not enough to guarantee the effect of the estimation.
c.	In Experiment Setting, the work follows the settings of the prior works to a very low amount of allowed budget. But the setting is not practical in real world. It will be better if the work can provide the performance on larger budget.
d.	In Baselines of 4.1, the work states it adopts k-means++ seeding algorithm as prior work. The ablation study on the effect of k-means++ seeding algorithm is needed.
e.	In Quantitative Analysis, the work analyzes the time complexity. The description of the equipment is needed.


**Summary Of The Paper:**

This work proposes to consider the generalization ability into the acquisition process of new labeled instances in active learning. The work selects instances with a high value of loss sharpness on pseudo-labels, which can be used for improving the generalization ability of the model based on the finding of Sharpness-Aware Minimization (SAM). Besides, the work theoretically gives the connection between the acquisition score and other scores, which are related to the generalization ability. In addition, the work shows consistent improvement over multiple datasets.

**Summary Of The Review:**

This work proposes to consider the generalization ability into the acquisition process of new labeled instances in active learning. The work selects instances with a high value of loss sharpness on pseudo-labels, which can be used for improving the generalization ability of the model based on the finding of Sharpness-Aware Minimization (SAM). Besides, the work theoretically gives the connection between the acquisition score and other scores, which are related to the generalization ability. In addition, the work shows consistent improvement over multiple datasets. However, the proposed method seems extend the SAM to active learning setting.

---

### Official Review · Reviewer_tmqo · 2022-10-24

**Confidence:** 4
**Correctness:** 3
**Technical Novelty And Significance:** 3
**Empirical Novelty And Significance:** 3
**Recommendation:** 6

**Clarity, Quality, Novelty And Reproducibility:**

Clarity: The paper is written nicely and is easy to follow.

Quality & Novelty: As discussed above, sufficient novelty is contained in the proposed method.

Reproducibility lacks sometimes.



**Strength And Weaknesses:**

Strength:
1. The paper is well written, the motivations for choices in the method are clear and the idea is simple yet effective, making full use of sharpness-aware minimization.
2. The presentation of background in Section 2 is thorough and complete.
3. The empirical evaluation is thorough and conducted on diverse vision-based tasks, including image classification and object detection.

Weakness:

[Motivation] The motivation is reasonable for me to consider the generalization ability in active learning. It might be better to give more examples to help understand why we need to select samples whose perturbed loss is maximum. For example, why should we apply SAM to active learning? Does SAM inherently have generalization properties?

[Relevant works] The work aims to connect active learning and generalization ability. Recent works about active domain adaptation also concern generalization. Several relevant references are missing [a, b, c, d]. The similarities and dissimilarities should be discussed.

[Experiment]
- For more clear understanding, could you provide some qualitative and quantitative results for the loss landscape?
- I would like to see why this work is not applied to the semantic segmentation problem and compared to works in that.

Refs:
[a] Active Domain Adaptation via Clustering Uncertainty-weighted Embeddings.

[b] Active Learning for Domain Adaptation: An Energy-Based Approach.

[c] Discrepancy-Based Active Learning for Domain Adaptation.

[d] Active Adversarial Domain Adaptation.

**Summary Of The Paper:**

This manuscript describes a new active learning method considering generalization, which is very practical. The key insight is to incorporate the sharpness of loss space when designing the active acquisition function and select unlabeled samples with the maximal perturbed loss for labeling. The experimental results and theoretical evidence confirm the effectiveness of the proposed approach.

**Summary Of The Review:**

Generally, I find this paper to touch on an interesting problem with a novel sharpness-aware minimization based active learning for generalization. Yet, it still requires clarification and some solid empirical support before warranting acceptance of this paper.

---

### Official Review · Reviewer_3NDR · 2022-10-24

**Confidence:** 3
**Correctness:** 3
**Technical Novelty And Significance:** 3
**Empirical Novelty And Significance:** 2
**Recommendation:** 6

**Clarity, Quality, Novelty And Reproducibility:**

a. The submission is clear and well-written.
b. The usage of sharpness with theoretical guarantees in active learning is interesting and new.
c. The submission does not provide source code. The experimental results are not guaranteed to have reproducibility.

**Strength And Weaknesses:**

Strengths:
a. The proposed method takes full advantage of sharpness information of loss.
b. Generalization error of active learning is upper bounded by the method.
c. Experiments verify the performance of the method.

Weakness:
a. More datasets should be considered in the experiments, especially for large-scale ones (e.g., Webvision, ImageNet).
b. In Table 1, the proposed method does not beat the previous studies with a much margin. It seems that SAM optimizer plays a more important role in the improvement.
c. The time complexity of the proposed method is much higher than that of the most of existing methods. Is it possible to accelerate the method without degradation?
d. Is there a clear cut of value for \rho to satisfy Proposition 3.2?


**Summary Of The Paper:**

The study investigates the active learning problem with sharpness information in loss space. It proposes a method named sharpness-aware active learning to select unlabeled instances potentially having maximal perturbation loss. The method has a theoretical guarantee on the generalization error of active learning. Experiment results show the effectiveness of the proposed method.

**Summary Of The Review:**

The work proposes a new method for active learning by adopting the sharpness information in loss space. The method is interesting with a design of minmax optimization. Experiments show the effectiveness of the method. However, the method does not outperform the existing methods, and the SAM optimizer affects more in the improvements. Also the proposed method is time consuming. Overall, the reviewer believes that the work is boarderline and needs further polishing.

---

### Decision · Program_Chairs · 2023-01-20

**Decision:**

Reject

**Justification For Why Not Higher Score:**

The authors did not respond to any of the concerns raised by reviewers. I am worried that this could be because they either found a flaw in their work, or felt the concerns were valid and could not be fixed without major revision.

**Justification For Why Not Lower Score:**

N/A

**Metareview: Summary, Strengths And Weaknesses:**

This paper proposes an new active learning algorithm. In active learning, since only a limited amount of data is selected by the algorithm for training, the chance of overfitting is higher than conventional batch learning. This work proposes a sample selection criterion that specifically is aimed to prevent overfitting. Specifically, the method utilizes the sharpness induced by a sample on the loss surface as the sample selection criterion, hence the name SAAL: Sharpness-Aware Active Learning. The idea is neat, sound, and simple. To the best of my knowledge, this is the first use case of sharpness aware optimization sample selection. The paper also presents a theoretical analysis of the method that provides an upper bound of SAAL  acquisition score and a connection to the leading eigenvalue of loss Hessian (an indicator of loss sharpness). in the context of active learning. Experiments are performed on image classification and object detection tasks, and demonstrate improvement over baseline methods. Initial reviews were borderline, but the authors chose not to reply to any of the reviews. As a result, the questions and concerns raised by reviewers are left unresolved, and unfortunately with that, the paper cannot be accepted.

I really find the idea interesting and I do hope authors consider to resubmit.